# COMPLEMENT OBJECTIVE TRAINING

**Hao-Yun Chen[1], Pei-Hsin Wang[1], Chun-Hao Liu[1], Shih-Chieh Chang[1, 2], Jia-Yu Pan[3], Yu-Ting Chen[3], Wei Wei[3], and Da-Cheng Juan[3]**

[1]Department of Computer Science, National Tsing-Hua University, Hsinchu, Taiwan
[2]Electronic and Optoelectronic System Research Laboratories, ITRI, Hsinchu, Taiwan
[3]Google Research, Mountain View, CA, USA
{haoyunchen,peihsin,newgod1992}@gapp.nthu.edu.tw
scchang@cs.nthu.edu.tw
{jypan, yutingchen, wewei, dacheng}@google.com

## ABSTRACT

Learning with a primary objective, such as softmax cross entropy for classification and sequence generation, has been the norm for training deep neural networks for years. Although being a widely-adopted approach, using cross entropy as the primary objective exploits mostly the information from the ground-truth class for maximizing data likelihood, and largely ignores information from the complement (incorrect) classes. We argue that, in addition to the primary objective, training also using a complement objective that leverages information from the complement classes can be effective in improving model performance. This motivates us to study a new training paradigm that maximizes the likelihood of the ground-truth class while neutralizing the probabilities of the complement classes. We conduct extensive experiments on multiple tasks ranging from computer vision to natural language understanding. The experimental results confirm that, compared to the conventional training with just one primary objective, training also with the complement objective further improves the performance of the state-of-the-art models across all tasks. In addition to the accuracy improvement, we also show that models trained with both primary and complement objectives are more robust to single-step adversarial attacks.

## 1 INTRODUCTION

Statistical learning algorithms work by optimizing towards a training objective. A dominant principle for training is to optimize likelihood (Mitchell et al., 1997), which measures the probability of data given the model under a specific set of parameters. The popularity of deep neural networks has given rise to the use of cross entropy (Kullback & Leibler, 1951) as its primary training objective, since minimizing cross entropy is essentially equivalent to maximizing likelihood for disjoint classes. Cross entropy has become the standard training objective for many tasks including classification (Krizhevsky et al., 2012) and sequence generation (Sutskever et al., 2014).

Let $\mathbf{y}_i \in \{0, 1\}^K$ be the label of the $i^{\text{th}}$ sample in one-hot encoded representation and $\hat{\mathbf{y}}_i \in [0, 1]^K$ be the predicted probabilities, the cross entropy $H(\mathbf{y}, \hat{\mathbf{y}})$ is defined as:

$$H(\mathbf{y}, \hat{\mathbf{y}}) = -\frac{1}{N} \sum_{i=1}^{N} \mathbf{y}_i^T \cdot \log(\hat{\mathbf{y}}_i)$$

$$= -\frac{1}{N} \sum_{i=1}^{N} \log(\hat{\mathbf{y}}_{ig}) \tag{1}$$

where $\hat{\mathbf{y}}_{ig}$ represents the predicted probability of the ground-truth class for the $i^{\text{th}}$ sample. Training with cross entropy as the primary objective aims at finding $\hat{\boldsymbol{\theta}} = \arg\min_{\boldsymbol{\theta}} H(\mathbf{y}, \hat{\mathbf{y}})$, where $\hat{\mathbf{y}} = h_{\boldsymbol{\theta}}(\mathbf{x})$, $h_{\boldsymbol{\theta}}$ is a neural network and $\mathbf{x}$ is a sample. Although training using the cross entropy as

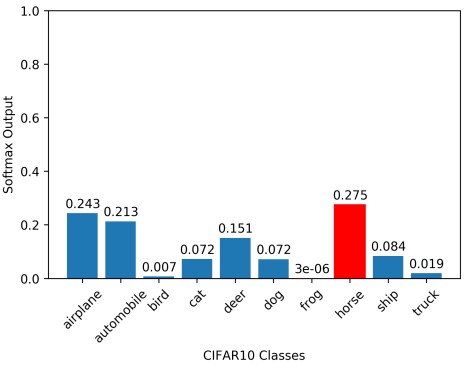 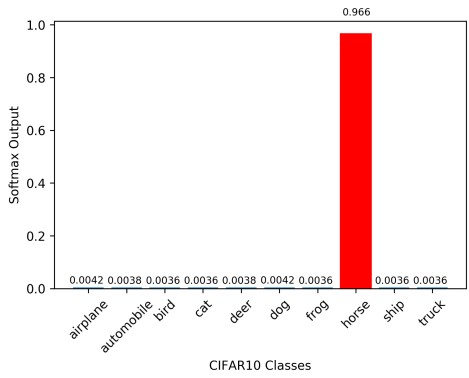

(a) $\hat{\mathbf{y}}$ from the model trained with cross entropy.

(b) $\hat{\mathbf{y}}$ from the model trained with COT.

Figure 1: Predicted probabilities $\hat{\mathbf{y}}$ from two training paradigms: (a) With cross entropy as the primary objective. (b) COT: with both primary and complement objectives. The model is ResNet-110 and the sample image is from CIFAR10 dataset. The ground-truth class is "horse." Compared to (b), the model in (a) is confused by other classes such as "airplane" and "automobile," which suggests (a) might be more susceptible for generalization issues and potentially adversarial attacks.

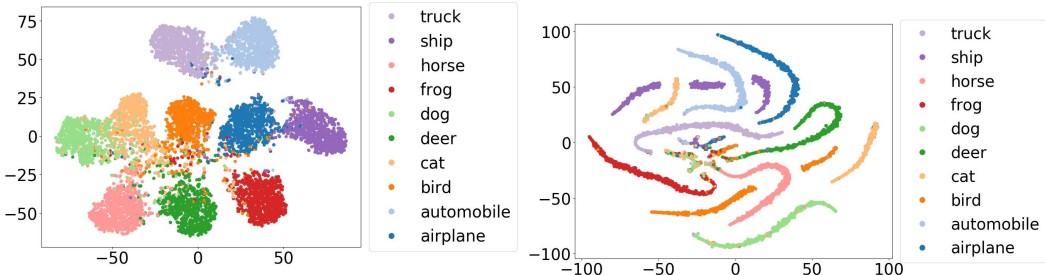

(a) Embeddings from the model trained with cross entropy.

(b) Embeddings from the model trained with COT.

Figure 2: Embeddings for CIFAR10 test images from two training paradigms: (a) With cross entropy as the primary objective. (b) COT: training with both primary and complement objectives. The model is ResNet-110, and the "embedding" is the vector representation before taking the softmax operation. The embedding representation of each sample is projected to two dimensions using t-SNE for visualization purpose. Compared to (a), the cluster of each class in (b) is "narrower" in terms of intra-cluster distance. Also, the clusters in (b) seem to have clean and separable boundaries, leading to more accurate and robust classification results.

the primary objective has achieved tremendous success, we have observed one limitation: it exploits mostly the information from the ground-truth class as Eq(1) shows; the information from *complement classes* (i.e., incorrect classes) has been largely ignored, since the predicted probabilities other than $\hat{\mathbf{y}}_{ig}$ are zeroed out due to the dot product calculation with the one-hot encoded $\mathbf{y}_i$. Therefore, for classes other than the ground truth, the model behavior is not explicitly optimized — their predicted probabilities are indirectly minimized when $\hat{\mathbf{y}}_{ig}$ is maximized since the probabilities sum up to 1. One way to utilize the information from the complement classes is to neutralize their predicted probabilities. To this end, we propose **C**omplement **O**bjective **T**raining (COT), a new training paradigm that achieves this optimization goal without compromising the model's primary objective. Figure 1 illustrates the comparison between Figure 1a: the predicted probability $\hat{\mathbf{y}}$ from the model trained with just cross entropy as the primary objective, and Figure 1b: $\hat{\mathbf{y}}$ from the model trained with both primary and complement objectives. Training with the complement objective finds the parameters $\boldsymbol{\theta}$ that evenly suppress complement classes without compromising the primary objective (i.e., maximizing $\hat{\mathbf{y}}_g$), making the model more confident of the ground-truth class.

Figure 2 further illustrates the embeddings of CIFAR10 images calculated from ResNet-110 using two training paradigms: cross entropy and COT. An embedding of an image is the vector representation computed by the ResNet-110 model, before taking the softmax operation. Compared to Figure 2a, the clusters in Figure 2b seem to have clean and separable boundaries, leading to more accurate and robust classification results. The experimental results later in Section 3 further confirm this observation.

Complement objective training requires a function that complements the primary objective. In this paper, we propose "complement entropy" (defined in Section 2) to complement the softmax cross entropy for neutralizing the effects of complement classes. The neural net parameters $\boldsymbol{\theta}$ are then updated by alternating iteratively between (a) minimizing cross entropy to increase $\hat{\mathbf{y}}_g$, and (b) maximizing complement entropy to neutralize $\hat{\mathbf{y}}_{j \neq g}$. Experimental results (in Section 3) confirm that COT improves the accuracies of the state-of-the-art methods for both (a) the image classification tasks on ImageNet-2012, Tiny ImageNet, CIFAR-10, CIFAR-100, and SVHN, and (b) language understanding tasks on machine translation and speech recognition. Furthermore, experimental results also show that models trained by COT are more robust to adversarial attacks.

## 2 Complement Objective Training

In this section, we first define "Complement Entropy" as the complement objective, and then provide a new training algorithm for updating neural network parameters $\boldsymbol{\theta}$ by alternating iteratively between the primary objective and the complement objective.

### 2.1 Complement entropy

Conventionally, training with cross entropy as the primary objective aims at maximizing the predicted probability of the ground-truth class $\hat{\mathbf{y}}_g$ in Eq(1). As mentioned in the introduction, the proposed COT also maximizes the complement objective for neutralizing the predicted probabilities of the complement classes. To achieve this, we propose "complement entropy" as the complement objective; complement entropy $C(\cdot)$ is defined to be the average of sample-wise entropies over complement classes in a mini-batch:

$$
\begin{aligned}
C(\hat{\mathbf{y}}_{\bar{c}}) &= \frac{1}{N} \sum_{i=1}^{N} \mathcal{H}(\hat{\mathbf{y}}_{i\bar{c}}) \\
&= -\frac{1}{N} \sum_{i=1}^{N} \sum_{j=1, j \neq g}^{K} \left(\frac{\hat{\mathbf{y}}_{ij}}{1 - \hat{\mathbf{y}}_{ig}}\right) \log\left(\frac{\hat{\mathbf{y}}_{ij}}{1 - \hat{\mathbf{y}}_{ig}}\right)
\end{aligned}
\tag{2}
$$

$\mathcal{H}(\cdot)$ is the entropy function. All the symbols and notations used in this paper are summarized in Table 1. One thing worth noticing is that this sample-wise entropy is calculated by considering only the complement classes other than the ground-truth class $g$. The sample-wise predicted probability $\hat{\mathbf{y}}_{ij}$ is normalized by one minus the ground-truth probability (i.e., $1 - \hat{\mathbf{y}}_{ig}$). The term $\hat{\mathbf{y}}_{ij}/(1 - \hat{\mathbf{y}}_{ig})$ can be understood as: conditioned on the ground-truth class $g$ not happening, the predicted probability to see the class $j$ for the $i^{\text{th}}$ sample. Since the entropy is maximized when the events are equally likely to occur, optimizing on the complement entropy drives $\hat{\mathbf{y}}_{ij}$ to $(1 - \hat{\mathbf{y}}_{ig})/(K - 1)$, which essentially neutralizes the predicted probability of complement classes as $K$ grows large. In other words, maximizing the complement entropy "flattens" the predicted probabilities of complement classes $\hat{\mathbf{y}}_{j \neq g}$. We conjecture that, when $\hat{\mathbf{y}}_{j \neq g}$ are neutralized, the neural net $h_{\boldsymbol{\theta}}$ generalizes better, since it is less likely to have an incorrect class with a sufficiently high predicted probability to "challenge" the ground-truth class.

### 2.2 Training with complement objective

Given a training procedure using a primary objective, such as softmax cross entropy, one can easily adopt the complement entropy to turn the procedure into a Complement Objective Training (COT). Algorithm 1 describes the new training mechanism by alternating iteratively between the primary and complement objectives. At each training step, the cross entropy is first calculated as the loss value to update the model parameters; next, the complement entropy is calculated as the loss value

to perform the second update. Therefore, additional forward and backward propagation are required in each iteration when using the complement objective, making the total training time empirically 1.6 times longer.

Table 1: Notations used in this paper.

| Symbol | Meaning |
|---|---|
| $\mathbf{y}_i$ | One-hot vector representing the label of the $i^{\text{th}}$ sample. |
| $\hat{\mathbf{y}}_i$ | The predicted probability for each class for the $i^{\text{th}}$ sample. |
| $g$ | Index of the ground-truth class. |
| $\mathbf{y}_{ij}$ or $\hat{\mathbf{y}}_{ij}$ | The $j^{\text{th}}$ class (element) of $\mathbf{y}_i$ or $\hat{\mathbf{y}}_i$. |
| $\hat{\mathbf{y}}_{\bar{c}}$ | Predicted probabilities of of the complement (incorrect) classes. |
| $H(\cdot, \cdot)$ | Cross entropy function. |
| $\mathcal{H}(\cdot)$ | Entropy function. |
| $C(\cdot)$ | Complement entropy. |
| $N$ and $K$ | Total number of samples and total number of classes. |

---

**Algorithm 1:** Training by alternating between primary and complement objectives

---
1 **for** $t \leftarrow 1$ **to** $n_{train\_steps}$ **do**
2 $\quad$ 1. Update parameters by Primary Objective: $-\frac{1}{N} \sum_{i=1}^{N} \log(\hat{\mathbf{y}}_{ig})$
3 $\quad$ 2. Update parameters by Complement Objective: $-\frac{1}{N} \sum_{i=1}^{N} \mathcal{H}(\hat{\mathbf{y}}_{i\bar{c}})$

---

## 3 EXPERIMENTS

We perform extensive experiments to evaluate COT on tasks in domains ranging from computer vision to natural language understanding and compare it with the baseline algorithms that achieve state-of-the-art in the respective domains. We also perform experiments to evaluate the robustness of the model trained by COT when attacked by adversarial examples. For each task, we select a state-of-the-art model that has an open-source implementation (referred to as "baseline") and reproduce their results with the hyper-parameters reported in the paper or code repository. Our code is available at https://github.com/henry8527/COT.

### 3.1 BALANCING TRAINING OBJECTIVES

In theory, the loss values between the primary and the complement objectives can be in different scales; therefore, additional efforts for tuning learning rates might be required for optimizers to achieve the best performance. Empirically, we find the complement entropy in Eq(2) can be modified as follows to balance the losses between the two objectives:

$$
\begin{aligned}
C'(\hat{\mathbf{y}}_{\bar{c}}) &= \frac{1}{K-1} \cdot C(\hat{\mathbf{y}}_{\bar{c}}) \\
&= \frac{1}{K-1} \cdot \frac{1}{N} \sum_{i=1}^{N} \mathcal{H}(\hat{\mathbf{y}}_{i\bar{c}})
\end{aligned}
\tag{3}
$$

where $K$ is the number of classes. This modification can be treated as the complement entropy $C(\cdot)$ being "normalized" by $(K-1)$. For all the experiments conducted in this paper, we use this normalized complement entropy as the complement objective to improve the baselines without further tuning of learning rates.

### 3.2 IMAGE CLASSIFICATION

We consider the following datasets for experiments with image classification: CIFAR-10, CIFAR-100, SVHN, Tiny ImageNet and ImageNet-2012. For CIFAR-10, CIFAR-100 and SVHN, we

choose the following baseline models: ResNet-110 (He et al., 2016b), PreAct ResNet-18 (He et al., 2016a), ResNeXt-29 (2×64d) (Xie et al., 2017), WideResNet-28-10 (Zagoruyko & Komodakis, 2016) and DenseNet-BC-121 (Huang et al., 2017b) with a growth rate of 32. For those five models, we use a consistent set of settings below, which is described in (He et al., 2016b). Specifically, the models are trained using SGD optimizer with momentum of 0.9. Weight decay is set to be 0.0001 and learning rate starts at 0.1, then being divided by 10 at the 100[th] and 150[th] epoch. The models are trained for 200 epochs, with mini-batches of size 128. The only exception here is for training WideResNet-28-10, we follow the settings described in (Zagoruyko & Komodakis, 2016), and the learning rate is divided by 10 at the 60[th], 120[th] and 180[th] epoch. In addition, no dropout (Srivastava et al., 2014) is applied to any baseline according to the best practices in (Ioffe & Szegedy, 2015). For Tiny ImageNet and ImageNet-2012, the baseline models are slightly different: we follow the settings from (Zhang et al., 2018), and the details are described in the corresponding paragraphs.

**CIFAR-10 and CIFAR-100.** CIFAR-10 and CIFAR-100 are datasets (Krizhevsky, 2009) that contain colored natural images of 32x32 pixels, in 10 and 100 classes, respectively. We follow the baseline settings (He et al., 2016b) to pre-process the datasets; both datasets are split into a training set with 50,000 samples and a testing set with 10,000 samples. During training, zero-padding, random cropping, and horizontal mirroring are applied to the images with a probability of 0.5. For the testing images, we use the original images of 32x32 pixels.

A comparison between the models trained using the primary objective and the COT model is illustrated in Figures 3a and 4a for CIFAR-10 and CIFAR-100 respectively. We show that COT consistently outperforms the baseline models. Some of the models, for example, ResNetXt-29, achieves a significant performance boost of 12.5% in terms of classification errors. For some other models such as WideResNet-28-10 and DenseNet-BC-121, the improvements are not as significant but are still large enough to justify the differences. Similar conclusions can be observed from the CIFAR-100 dataset. In addition to the comparisons of the performance, we also present the change of testing errors over the course of the training in Figures 3b and 4b for the ResNet-110 model. Following the standard training practice, learning rates drop after the 100[th] epoch, which corresponds to a drop in testing errors. As we can see from the plot, COT outperforms consistently compared to the baseline models when the models are close to the convergence.

**Street View House Numbers (SVHN).** The SVHN dataset (Netzer et al., 2011) consists of images extracted from Google Street View. We divide the dataset into a set of 73,257 digits for training and a set of 26,032 digits for testing. When pre-processing the training and validation images, we follow the general practice to normalize pixel values into [-1,1]. Table 2 shows the experimental results and confirms that COT consistently improves the baseline models with the biggest improvement being the ResNet-110 with 11.7% reduction on the error rate.

| Model | Baseline | COT |
|---|---|---|
| ResNet-110 | 7.56 | **6.84 (6.99±0.12)** |
| PreAct ResNet-18 | 5.46 | **4.86 (5.08±0.14)** |
| ResNeXt-29 (2×64d) | 5.20 | **4.55 (4.69±0.12)** |
| WideResNet-28-10 | 4.40 | **4.30 (4.34±0.03)** |
| DenseNet-BC-121 | 4.72 | **4.62 (4.67±0.03)** |

(a) Test errors (in %) on CIFAR-10. For COT, we repeat 5 runs and report the "best (mean±std)" error values.

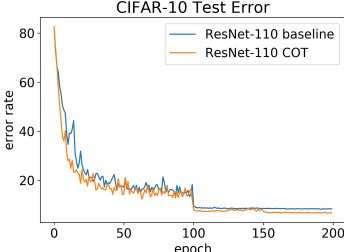

(b) Test errors of ResNet-110 on CIFAR-10 over epochs.

Figure 3: Classification errors on CIFAR-10: (a) COT improves all 5 state-of-the-art models. (b) The improvement over epochs. Notice that the performance improvement from COT becomes stable after the 100[th] epoch due to the learning rate decrease.

| Model | Baseline | **COT** |
|---|---|---|
| ResNet-110 | 29.22 | **27.90** |
| PreAct ResNet-18 | 25.44 | **24.73** |
| ResNeXt-29 (2×64d) | 23.45 | **21.90** |
| WideResNet-28-10 | 21.91 | **20.99** |
| DenseNet-BC-121 | 21.73 | **20.54** |

(a) Test errors (in %) on CIFAR-100. For COT, we repeat 3 runs and report the mean value.

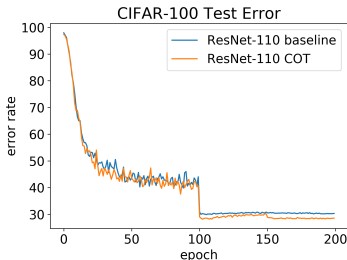

(b) Test errors of ResNet-110 on CIFAR-100 over the epochs.

Figure 4: Classification errors on CIFAR-100: (a) COT improves all 5 state-of-the-art models. (b) The improvement over epochs. Similar to the trend observed in CIFAR-10, the performance improvement from COT becomes stable after the 100[th] epoch due to the learning rate decrease.

Table 2: Test errors (in %) of the baseline models and the COT-trained models on the SVHN dataset. The values presented are the mean values of 3 runs.

| Model | Baseline | **COT** |
|---|---|---|
| ResNet-110 | 4.94 | **4.36** |
| PreAct ResNet-18 | 4.31 | **3.96** |
| ResNeXt-29 (2×64d) | 4.22 | **3.76** |
| WideResNet-28-10 | 3.72 | **3.50** |
| DenseNet-BC-121 | 3.52 | **3.47** |

**Tiny ImageNet.** Tiny ImageNet[1] dataset is a subset of ImageNet (Deng et al., 2009), which contains 100,000 images for training and 10,000 for testing images across 200 classes. In this dataset, each image is down-sampled to 64x64 pixels from the original 256x256 pixels. We consider four state-of-the-art models as baselines: ResNet-50, ResNet-101 (He et al., 2016b), ResNeXt-50 (32×4d) and ResNeXt-101 (32×4d) (Xie et al., 2017). During training, we follow the standard data-augmentation techniques, such as random cropping, horizontal flipping, and normalization. For each model, the stride of the first convolution layer is modified to adapt images of size 64x64 (Huang et al., 2017a). For evaluation, the testing data is only augmented with 56x56 central cropping. The rest of the experimental details are the same as the ones described at the beginning of Section 3.2. Table 3 provides the experimental results, which demonstrate that COT consistently improves the performance of all baseline models.

Table 3: Top-1 Validation errors (in %) for the Tiny ImageNet experiments (mean values of 3 runs).

| Model | Baseline | **COT** |
|---|---|---|
| ResNet-50 | 39.39 | **39.20** |
| ResNet-101 | 38.23 | **37.35** |
| ResNeXt-50 (32×4d) | 37.36 | **36.69** |
| ResNeXt-101 (32×4d) | 37.02 | **36.14** |

**ImageNet.** ImageNet-2012 dataset (Russakovsky et al., 2015) is one of the largest datasets for image classification, which contains 1.3 million images for training and 50,000 images for testing with 1,000 classes. Random crops and horizontal flips are applied during training (He et al., 2016b), while images in the testing set use 224x224 center crops (1-crop testing) for data augmentation. ResNet-50 is selected as the baseline model, and we follow (Goyal et al., 2017) for the experimental

---

[1]https://tiny-imagenet.herokuapp.com/

setup: 256 minibatch size, 90 total training epochs, and 0.1 as the initial learning rate starting that is decayed by dividing 10 at the $30^{th}$, $60^{th}$ and $80^{th}$ epoch. Table 4 shows (a) the error rate[2] of baseline reported by (He et al., 2016b) and (b) the error rate of baseline model trained by COT, which confirms COT further improves the baseline performance.

Table 4: Validation errors (in %) for the ImageNet-2012 experiments.

| Model | | Baseline | **COT** |
|---|---|---|---|
| ResNet-50 | Top-1 Error | 24.7 | **24.4** |

## 3.3 NATURAL LANGUAGE UNDERSTANDING

COT is also evaluated on two natural language understanding (NLU) tasks: machine translation and speech recognition. One distinct characteristic of most NLU tasks is a large number of target classes. For example, the machine translation dataset used in this paper, IWSLT 2015 English-Vietnamese (Cettolo et al., 2015), consists of vocabularies of 17,191 English words and 7,709 Vietnamese words. This necessitates the normalized complement entropy in Eq(3).

**Machine translation.** Neural machine translation (NMT) has popularized the use of neural sequence models (Sutskever et al., 2014; Cho et al., 2014). Specifically, we apply COT on the seq2seq model with Luong attention mechanism (Luong et al., 2015) on the IWSLT 2015 English-Vietnamese dataset, which contains 133 thousand translation pairs. For validation and testing, we use TED tst2012 and TED tst2013, respectively. For the baseline implementation, we follow the official TensorFlow-NMT implementation[3]. That is, the number of total training steps is 12,000 and the weight decay starts at the $8,000^{th}$ step then applied for every 1,000 steps. We experiment models with both greedy decoder and beam search decoder. The model trained by COT gives the best testing results when the beam width is 3, while the baseline uses 10 as the best beam width. Table 5 illustrates the experimental results, showing COT improves testing BLEU scores compared to the baseline NMT model on both greedy decoder and the beam search decoder.

Table 5: Results of IWSLT 2015 English-Vietnamese. The BLEU scores on tst2013 are reported.

| Model | Baseline | **COT** |
|---|---|---|
| NMT (greedy) | 25.5 | **25.7** |
| NMT (beam search) | 26.1 | **26.4** |

**Speech recognition.** For speech recognition, we experiment on Google Commands Dataset (Warden, 2018), which consists of 65,000 one-second utterances of 30 different types such as "Yes," "No," "Up," "Down" and "Stop." Our baseline model is referenced from (Zhang et al., 2018). We apply the same pre-processing steps as shown in the paper, and perform the short-time Fourier transform on the original waveforms first at a sampling rate of 4 kHz to receive the corresponding spectrograms. We then zero-pad these spectrograms to equalize each sample's length. For the baseline model, we select VGG-11 (Simonyan & Zisserman, 2014) and train the model for 30 epochs following the steps in (Zhang et al., 2018). We use SGD optimizer with momentum, and weight decay is 0.0001. The learning rate starts at 0.0001 and then is divided by 10 at the $10^{th}$ and $20^{th}$ epoch. COT improves the baseline by further reducing the error rate by 1.56%, as shown in Table 6.

## 3.4 ADVERSARIAL EXAMPLES

An adversarial example is an imperceptibly-perturbed input that results in the model outputting an incorrect answer with high confidence (Szegedy et al., 2014; Goodfellow et al., 2015). Prior

---

[2] https://github.com/KaimingHe/deep-residual-networks
[3] https://github.com/tensorflow/nmt/tree/tf-1.4

Table 6: Test errors (in %) on Google Commands Dataset (mean values of 3 runs).

| Model | Baseline | **COT** |
|-------|----------|---------|
| VGG-11 | 6.06 | **4.50** |

literatures have shown that there are several methods to generate effective adversarial examples that greatly mislead the model toward providing wrong predictions.

As shown in Figure 2, the proposed COT generates embeddings where the class boundaries are clear and well-separated. We believe that the models trained using COT generalize better and are more robust to adversarial attacks. To verify this conjecture, we conduct experiments of white-box attacks to the models trained by COT. We consider a common approach of single-step adversarial attacks: Fast Gradient Sign Method (FGSM) (Goodfellow et al., 2015) that uses the gradient to determine the direction of the perturbation to apply on an input for creating an adversarial example. To set up FGSM white-box attacks on a baseline model, adversarial examples are generated using the gradients calculated based on the primary objective (referred to as the "primary gradient") of the baseline model. For FGSM white-box attacks on COT, adversarial perturbations are generated based on the sum of the primary gradient and the complement gradient (*i.e.*, the gradient calculated from the complement objective), both gradients from the model trained by COT. In our experiments, the baseline models are the same as in Section 3.2, and the amount of perturbation is limited to a maximum value of 0.1 as described in (Goodfellow et al., 2015) when creating adversarial examples. Furthermore, we also conduct experiments on FGSM transfer attacks, which use the adversarial examples from a baseline model to attack and test the robustness of the model trained by COT.

Table 7: Classification errors (in %) on CIFAR-10 under FGSM white-box & transfer attacks.

| Model | Baseline | **COT (White-Box)** | **COT (Transfer)** |
|-------|----------|---------------------|--------------------|
| ResNet-110 | 62.23 | **52.72** | **54.96** |
| PreAct ResNet-18 | 65.60 | **56.17** | **59.39** |
| ResNeXt-29 (2×64d) | 70.24 | **61.55** | **65.83** |
| WideResNet-28-10 | 59.39 | **55.53** | **57.33** |
| DenseNet-BC-121 | 65.97 | **55.99** | **62.40** |

Table 7 shows the performance of the models on the CIFAR-10 dataset under FGSM white-box and transfer attacks. Generally, the models trained using COT have lower classification error under both FGSM white-box and transfer attacks, which is an indicator that COT models are more robust to both kinds of attacks. We also conduct experiments on the basic iterative attacks using I-FGSM (Kurakin et al., 2017) and the corresponding results can be found in Appendix A.

We conjecture that since the main goal of the complement gradients is to neutralize the probabilities of incorrect classes (instead of maximizing the probability of the correct class), the complement gradients may "push away" primary gradients when forming adversarial perturbations, which might partially answer why COT is more robust to FGSM white-box attacks compared to the baseline. Regarding the transfer attacks, only the primary objective of the baseline model is used to calculate the gradients for generating adversarial examples. In other words, the complement gradients are not considered when generating adversarial examples in the transfer attack, and this might be the reason why models trained by COT are more robust to transfer attacks. Both conjectures leave a large space for future work: using complement objective to defend against more advanced adversarial attacks.

## 4    CONCLUSION AND FUTURE WORK

In this paper, we study Complement Objective Training (COT), a new training paradigm that optimizes the complement objective in addition to the primary objective. We propose complement entropy as the complement objective for neutralizing the effects of complement (incorrect) classes.

Models trained using COT demonstrate superior performance compared to the baseline models. We also find that COT makes the models robust to single-step adversarial attacks.

COT can be extended in several ways: first, in this paper, the complement objective is chosen to be the complement entropy. Non-entropy-based complement objectives should also be considered for future studies, which is left as a straight-line future work. Secondly, the exploration of COT on broader applications remains as an open research question. One example would be applying COT on generative models such as Generative Adversarial Networks (Goodfellow et al., 2014). Another example would be using COT on object detection and segmentation. Finally, in this work, we show using complement objective help defend single-step adversarial attacks; the behavior of COT on more advanced adversarial attacks deserves further investigation and is left as another future work.

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

# A    ITERATIVE FAST GRADIENT SIGN METHOD

Table 8: Classification errors (in %) on CIFAR-10 under I-FGSM transfer attacks.

| Model | Baseline | **COT (Transfer)** |
|---|---|---|
| ResNet-110 | 88.00 | **84.36** |
| PreAct ResNet-18 | 84.56 | **83.77** |
| ResNeXt-29 (2×64d) | 87.79 | **86.43** |
| WideResNet-28-10 | 81.97 | **80.85** |
| DenseNet-BC-121 | 86.95 | **83.66** |

Table 8 shows the performance of the models on the CIFAR-10 dataset under I-FGSM transfer attacks. Generally, the models trained using COT have lower classification error under I-FGSM transfer attacks. The number of iteration is set to 10 in the experiment.

