# OpenReview forum: "Complement Objective Training"
_ICLR.cc/2019/Conference_

### Official Review · AnonReviewer3 · 2018-10-31
**Interesting new idea, good experimental results, some points to clarify.**

**Rating:** 7
**Confidence:** 4

**Review:**


========
Summary
========

The paper deals with the training of neural networks for classification or sequence generation tasks, using a cross-entropy loss. Minimizing the cross-entropy means maximizing the predicted probabilities of the ground-truth classes (averaged over the samples). The authors introduce a "complementary entropy" loss with the goal of minimizing the predicted probabilities of the complementary (incorrect) classes. To do that, they use the average of sample-wise entropy over the complement classes. By maximizing this entropy, the predicted complementary probabilities are encouraged to be equal and therefore, the authors claim that it neutralizes them as the number of classes grows large. The proposed training procedure, named COT, consists of alternating between the optimization of the two losses.

The procedure is tested on image classification tasks with different datasets (CIFAR-10, CIFAR-100, Street View House Numbers, Tiny ImageNet and ImageNet), machine translation (training using IWSLT dataset, validation and test using TED tst2012/2013 datasets), and speech recognition (Gooogle Commands dataset). In the experiments, COT outperforms state-of-the-art models for each task/dataset.

Adversarial attacks are also considered for the classification of images of CIFAR-10: using the Fast Gradient Sign and Basic Iterative Fast Gradient Sign methods on different models, adversarial examples specifically designed for each model, are generated. Then results of these models are compared to COT on these examples. The authors admit
that the results are biased since the adversarial attacks only target part of the COT objective, hence more accurate comparisons should be done in future work.

===========================
 Main comments and questions
===========================

End of page 1: "the model behavior for classes other than the ground  truth stays unharnessed and not well-defined". The probabilities  should still sum up to 1, so if the ground truth one is maximized,  the others are actually implicitly minimized. No?

Page 3, sec 2.1: "optimizing on the complement entropy drives ŷ_ij to 1/(K − 1)". I believe that it drives each term ŷ_ij /(1 − ŷ_ig ) to be equal to 1/(K-1). Therefore, it drives ŷ_ij to (1 − ŷ_ig)/(K-1) for j!=g.

This indeed flattens the ŷ_ij for j!=g, but the effect on ŷ_ig is not controlled. In particular this latter can decrease. Then in the next step of the algorithm, ŷ_ig will be maximized, but with no explicit control over the complementary probabilities. There are two objectives that are optimized over the same variable theta. So the question is, are we sure that this procedure will converge? What prevents situations where the probabilities will alternate between two values?

For example, with 4 classes, we look at the predicted probabilities of a given sample of class 1:
Suppose after step 1 of Algo 1, the predicted probabilities are:  0.5 0.3 0.1 0.1
After step 2:  0.1 0.3 0.3 0.3
Then step 1: 0.5 0.3 0.1 0.1
Then step 2: 0.1 0.3 0.3 0.3
And so on... Can this happen? Or why not? Did the algorithm have trouble converging in any of the experiments?

Sec 3.1:
"additional efforts for tuning hyper-parameters might be required for optimizers to achieve the best performance": Which hyper-parameters are considered here? If it is the learning rate, why not use a different one, tuned for each objective?

Sec 3.2:
The additional optimization makes each training iteration more costly. How much more? How do the total running times of COT compare to the ones of the baselines? I think this should be mentioned in the paper.

Sec 3.4:
As the authors mention, the results are biased and so the comparison is not fair here. Therefore I wonder about the  relevance of this section. Isn't there an easy way to adapt the attacks to the two objectives to be able to illustrate the conjectured robustness of COT? For example, naively having a two steps perturbation of the input: one based on the gradient of the primary objective and then perturb the result using the gradient of the complementary objective?

===========================
Secondary comments and typos
===========================

Page 3, sec 2.1: "...the proposed COT also optimizes the complement objective for neutralizing the predicted probabilities...", using maximizes instead of optimizes would be clearer.

In the definition of the complement entropy, equation (2), C takes as parameter only y^hat_Cbar but then in the formula, ŷ_ig appears. Shouldn't C take all \hat_y as an argument in this case?

Algorithm 1 page 4: I find it confusing that the (artificial) variable that appears in the argmin (resp. argmax) is theta_{t-1}
(resp. theta'_t) which is the previous parameter. Is there a reason for this choice?

Sec 3:
"We perform extensive experiments to evaluate COT on the tasks" --> COT on tasks

"compare it with the baseline algorithms that achieve state-of-the-art in the respective domain." --> domainS

"to evaluate the model’s robustness trained by COT when attacked" needs reformulation.

"we select a state- of-the-art model that has the open-source implementation" --> an open-source implementation

Sec 3.2:
Figure 4: why is the median reported and not the mean (as in Figure 3, Tables 2 and 3)?

Table 3 and 4: why is it the validation error that is reported and not the test error?

Sec 3.3:
"Neural machine translation (NMT) has populated the use of neural sequence models": populated has not the intended meaning.

"We apply the same pre-processing steps as shown in the model" --> in the paper?

Sec 3.4:
"We believe that the models trained using COT are generalized better" --> "..using COT generalize better"

"using both FGSM and I-FGSM method" --> methodS

"The baseline models are the same as Section 3.2." --> as in Section 3.2.

"the number of iteration is set at 10." --> to 10

"using complement objective may help defend adversarial attacks." --> defend against

"Studying on COT and adversarial attacks.." --> could be better formulated

References: there are some inconsistencies (e.g.: initials versus first name)


Pros
====
- Paper is clear and well-written
- It seems to me that it is a new original idea
- Wide applicability
- Extensive convincing experimental results

Cons
====
- No theoretical guarantee that the procedure should converge
- The training time may be twice longer (to clarify)
- The adversarial section, as it is,  does not seem relevant for me

---

> ### Author Response · Authors · 2018-11-14
> **Response to AnonReviewer3 [2/2]**
>
>
> (Q5) Sec 3.4: As the authors mention, the results are biased and so the comparison is not fair here. Therefore I wonder about the  relevance of this section. Isn't there an easy way to adapt the attacks to the two objectives to be able to illustrate the conjectured robustness of COT? For example, naively having a two steps perturbation of the input: one based on the gradient of the primary objective and then perturb the result using the gradient of the complementary objective?
>
> (A5) Thanks for the comment. We should have made clear that “black box” [1] (rather than “white box”) adversarial attacks are considered in the manuscript. Specifically, we follow the common practice of generating adversarial examples using both FGSM and I-FGSM methods with the gradients from a baseline model; this way, the model trained by COT is actually a “black box” to these attacks. We have modified the manuscript to clarify this part. Also, thanks for the great suggestion of forming adversarial attacks using “both” gradients (from both primary & complement objectives). We are designing and conducting experiments at the moment and will share results when ready.
>
>
> For the part of secondary comments and typos, we appreciate your thorough reading again and have corrected all these typos according to your suggestions. Meanwhile, in the following, we also provided explanations to your secondary comments.
>
>
> (Q1) Page 3, sec 2.1: "...the proposed COT also optimizes the complement objective for neutralizing the predicted probabilities...", using maximizes instead of optimizes would be clearer.
>
> (A1) Thanks for the suggestion. We have reworded the manuscript to “maximizes.”
>
>
> (Q2) In the definition of the complement entropy, equation (2), C takes as parameter only y^hat_Cbar but then in the formula, ŷ_ig appears. Shouldn't C take all \hat_y as an argument in this case?
>
> (A2) Since the probabilities sum up to one, ŷ_ig can be inferred from y^hat_Cbar. Also, for us, it seems more direct and clear to show that complement entropy is calculated from y^hat_Cbar when C takes y^hat_Cbar as the only argument. Therefore, we incline to keep the orignal formulation. If the reviewer has strong preference, please kindly let us know and we are happy to make changes accordingly.
>
>
> (Q3) Algorithm 1 page 4: I find it confusing that the (artificial) variable that appears in the argmin (resp. argmax) is theta_{t-1}
> (resp. theta'_t) which is the previous parameter. Is there a reason for this choice?
>
> (A3) Thanks for the comment. Originally, we want to notify readers that there are two backprops within one iteration. We agree that those symbols are confusing and therefore we have modified the manuscript with those symbols removed.
>
>
> (Q4) Sec 3.2  Figure 4: why is the median reported and not the mean (as in Figure 3, Tables 2 and 3)?
>
> (A4) Thanks for pointing this out. This is a typo and we have already corrected it in the manuscript: median -> mean.
>
>
> (Q5) Sec 3.2, Table 3 and 4: why is it the validation error that is reported and not the test error?
>
> (A5) Thanks for the detailed comment. For a fair comparison, we report the error in the exact same way as the open-sourced repo from the ResNet authors:
> https://github.com/KaimingHe/deep-residual-networks.
>
>
> (Q6) Sec 3.3: "Neural machine translation (NMT) has populated the use of neural sequence models": populated has not the intended meaning.
>
> (A6) We thank the reviewer for pointing out this typo. We have already corrected it in our manuscript: populated -> popularized
>
>
> (Q7) "Studying on COT and adversarial attacks.." --> could be better formulated
>
> (A7) Thanks for the comment again. We have modified the manuscript as follows: "Studying on the relationship between COT and adversarial attacks…”
>
>
> [1] Hongyi Zhang, Moustapha Cisse, Yann N. Dauphin, David Lopez-Paz. “Mixup: Beyond Empirical Risk Minimization.” In International Conference on Learning Representation, 2018.

---

> ### Author Response · Authors · 2018-11-14
> **Response to AnonReviewer3 [1/2]**
>
>
> We sincerely thank the reviewer for the useful and detailed comments. Below we provide explanations for each of your comments or questions.
>
>
> (Q1) End of page 1: "the model behavior for classes other than the ground truth stays unharnessed and not well-defined". The probabilities should still sum up to 1, so if the ground truth one is maximized,  the others are actually implicitly minimized. No?
>
> (A1) Your understanding is totally correct. We have changed the original text to a more clear statement:
>
> “Therefore, for classes other than the ground truth, the model behavior is not explicitly optimized --- their predicted probabilities are indirectly minimized when ŷ_ig is maximized since the probabilities sum up to 1.”
>
> We want to thank the reviewer again for crystalizing the manuscript.
>
>
> (Q2) Page 3, sec 2.1: "optimizing on the complement entropy drives ŷ_ij to 1/(K − 1)". I believe that it drives each term ŷ_ij /(1 − ŷ_ig ) to be equal to 1/(K-1). Therefore, it drives ŷ_ij to (1 − ŷ_ig)/(K-1) for j!=g.
>
> This indeed flattens the ŷ_ij for j!=g, but the effect on ŷ_ig is not controlled. In particular this latter can decrease. Then in the next step of the algorithm, ŷ_ig will be maximized, but with no explicit control over the complementary probabilities. There are two objectives that are optimized over the same variable theta. So the question is, are we sure that this procedure will converge? What prevents situations where the probabilities will alternate between two values?
>
> For example, with 4 classes, we look at the predicted probabilities of a given sample of class 1:
> Suppose after step 1 of Algo 1, the predicted probabilities are:  0.5 0.3 0.1 0.1
> After step 2:  0.1 0.3 0.3 0.3
> Then step 1: 0.5 0.3 0.1 0.1
> Then step 2: 0.1 0.3 0.3 0.3
> And so on... Can this happen? Or why not? Did the algorithm have trouble converging in any of the experiments?
>
> (A2) Thanks for the detailed comment. As the reviewer pointed out, “drives ŷ_ij to 1/(K − 1)” was indeed a typo and should be corrected to “drive ŷ_ij /(1 − ŷ_ig) to 1/(K-1)”. We have modified the manuscript correspondingly. Indeed, maximizing complement entropy in Eq(2) only drives “ŷ_ij /(1 − ŷ_ig) to 1/(K-1)”, and therefore in the example provided above, the predicted probabilities after step 2 can be “0.1 0.3 0.3 0.3” or “0.5, (1 - 0.5)/3, (1 - 0.5)/3, (1 - 0.5)/3”, or other values so long as the incorrect classes (ŷ_ij's) receive similar predicted probabilities. According to our observations from the experiments, the probabilities tend to converge to “0.5, (1 - 0.5)/3, (1 - 0.5)/3, (1 - 0.5)/3”. Experiments show that the algorithm does not have trouble converging; the algorithm converges smoothly in all the experiments we have conducted. Again, we thank the reviewer for the insightful comment; studying the theory of COT convergence is an intriguing topic and we leave it as a future work.
>
>
> (Q3) Sec 3.1: "additional efforts for tuning hyper-parameters might be required for optimizers to achieve the best performance": Which hyper-parameters are considered here? If it is the learning rate, why not use a different one, tuned for each objective?
>
> (A3) Hyper-parameters in this statement indeed refer to the learning rate, and we have modified the statement in the manuscript to avoid confusion; the modified statement is provided below:
>
> “therefore, additional efforts for tuning learning rates might be required for optimizers to achieve the best performance.”
>
> Regarding the second question about tuning learning rates, we have conducted several experiments with different learning rates specifically tuned for each objective. The experimental results show that using the same learning rate for both primary and complement objectives leads to the best performance when Eq(3) is used as the complement objective.
>
>
> (Q4) Sec 3.2: The additional optimization makes each training iteration more costly. How much more? How do the total running times of COT compare to the ones of the baselines? I think this should be mentioned in the paper.
>
> (A4) Yes, one additional backpropagation is required in each iteration when applying COT. On average, the total training time is about 1.6 times longer compared to the baselines. Thanks for the suggestion, and we have included this in the latest manuscript (section 2.2).

---

### Official Review · AnonReviewer2 · 2018-11-05
**Simple and sensible heuristic with impressive improvement**

**Rating:** 8
**Confidence:** 4

**Review:**

This paper considers augmenting the cross-entropy objective with "complement" objective maximization, which aims at neutralizing the predicted probabilities of classes other than the ground truth one. The main idea is to help the ground truth label stands out more easily by smoothing out potential peaks in non-ground-truth labels. The wide application of the cross-entropy objective makes this approach applicable to many different machine/deep learning applications varying from computer vision to NLP.

The paper is well-written, with a clear explanation for the motivation of introducing the complement entropy objective and several good visualization of its empirical effects (e.g., Figures 1 and 2). The numerical experiments also incorporate a wide spectrum of applications and network structures as well as dataset sizes, and the performance improvement is quite impressive and consistent. In particular, the adversarial attacks example looks very interesting.

One small suggestion is that the authors can also make some comments on the connection between the two-step update algorithm (Algorithm 1) with multi-objective optimization. In particular, I would suggest the authors also try some multi-objective optimization techniques apart from the simple but effective heuristics, and see if some Pareto-optimality can be guaranteed and better practical improvement can be achieved.

---

> ### Author Response · Authors · 2018-11-14
> **Response to AnonReviewer2**
>
>
> (Q1) One small suggestion is that the authors can also make some comments on the connection between the two-step update algorithm (Algorithm 1) with multi-objective optimization. In particular, I would suggest the authors also try some multi-objective optimization techniques apart from the simple but effective heuristics, and see if some Pareto-optimality can be guaranteed and better practical improvement can be achieved.
>
> (A1) We sincerely thank the reviewer for the helpful and constructive suggestion about associating COT with multi-objective optimization. This is really a brilliant idea. As a straight-line future work, we will survey multi-objective optimization techniques, and explore the direction of formulating COT into a multi-objective optimization problem.

---

### Official Review · AnonReviewer1 · 2018-11-05
**Nice idea but leaves several questions not answered**

**Rating:** 5
**Confidence:** 4

**Review:**

In this manuscript, the authors propose a secondary objective for softmax minimization. This complementary objective is based on evaluating the information gathered from the incorrect classes. Considering these two objectives leads to a new training approach. The manuscript ends with a collection of tests on a variety of problems.

This is an interesting point of view but the manuscript lacks discussion on several important questions:

1) How is this idea related to regularization? If we increase the regularization parameter, we can attain sparse parameter vectors.
2) Would this method also complement from overfitting?
3) In the numerical experiments, the comparison is carried out against a "baseline" method. Do the authors use regularization with these baseline methods? I believe the comparison will be fair  if the regularization option is turned on for the baseline methods.
4) Why combining the two objectives in a single optimization problem and then solving the resulting problem is not an option instead of the alternating method given in Algorithm 1?
5) How does alternating between two objectives change the training time? Do the authors use backpropagation?

---

> ### Author Response · Authors · 2018-11-14
> **Response to AnonReviewer1 [2/2]**
>
>
> (Q4) Why combining the two objectives in a single optimization problem and then solving the resulting problem is not an option instead of the alternating method given in Algorithm 1?
>
> (A4) We are very grateful for this novel idea, and we have conducted several preliminary experiments to explore this idea. Below are the comparisons between (a) the original COT method, and (b) the approach of combining the two objectives into one single objective. The experimental results show that the original COT method works better in almost all cases, and we conjecture that these two methods converge to different local minima. This idea is worth exploring, and we leave it as a straight-line future work.
>
> Test error of the state-of-the-art architectures on Cifar10
> ===========================================================
> 				                Combining into one objective	  COT
> ResNet-110 			        7.42%                 		                  6.84%
> PreAct ResNet-18                 4.92%                 		                  4.86%
> ResNeXt-29 (2×64d)             4.79%                 		                  4.55%
> WideResNet-28-10		4.00%                 		                  4.30%
> DenseNet-BC-121           	4.64%                 		                  4.62%
> ===========================================================
>
> Test error of the state-of-the-art architectures on Cifar100
> ===========================================================
> 				                 Combining into one objective	  COT
> ResNet-110 			         28.80%                 		                  27.90%
> PreAct ResNet-18                  25.30%                 		                  24.73%
> ResNeXt-29 (2×64d)              23.20%                 		                  21.90%
> WideResNet-28-10		 21.96%                 		                  20.99%
> DenseNet-BC-121           	 22.17%                 		                  20.54%
> ===========================================================
>
>
> (Q5) How does alternating between two objectives change the training time? Do the authors use backpropagation?
>
> (A5) Yes, we do use backpropagation. One additional backpropagation is required in each iteration when applying COT, and therefore the overall training time is about 1.6 times longer according to our experiments.
>
>
> [1] Kaiming He, Xiangyu Zhang, Shaoqing Ren, Jian Sun. “Deep Residual Learning for Image Recognition.” In IEEE Conference on Computer Vision and Pattern Recognition, 2016.
> [2] Sergey Zagoruyko, Nikos Komodakis. “Wide Residual Networks
> .” In British Machine Vision Conference, 2016.
> [3] Gao Huang, Zhuang Liu, Laurens van der Maaten, Kilian Q. Weinberger, David Lopez-Paz. “Densely Connected Convolutional Networks
> .” In IEEE Conference on Computer Vision and Pattern Recognition, 2017.

---

> > ### Comment · AnonReviewer1 · 2018-11-23
> > **Merging the objectives**
> >
> > This is interesting. From the response of the authors, I presume that the authors have simply added the two objectives together. However, it is more common to merge multiple objectives by premultiplying them with some weights. Since there are only two objectives, these two weights could be set with some kind of grid search (maybe along with cross-validation). I believe the tables given in the response would then change and the training times would decrease.
> >
> > Please report the increase in training time in the manuscript.

---

> > > ### Author Response · Authors · 2018-11-27
> > > **Follow-up**
> > >
> > > Thank you for the ideas. Yes, we indeed directly added the two objectives together in our experiments. We agree that introducing two additional weights to merge the primary and complement objectives is a good idea, and with proper tuning, this approach may further improve the model's performance and reduce the training time. We aimed to design a methodology with fewer hyper-parameters, so we didn't explore this direction, and our current proposed method works in many scenarios, as shown in our experiments.  With these promising results, we will continue to explore the approach of merging the two objectives, and build connections between these two approaches, in our immediate future work.
> > >
> > > Regarding reporting the increase in training time, we have added the information of training time in section 2.2 (on the top of page 4).

---

> > > > ### Comment · Area_Chair1 · 2018-12-10
> > > > **Which objectives were added together?**
> > > >
> > > > Did you sum the cross-entropy with the complement entropy (Eq. 2) or the normalized complement entropy (Eq. 3)?

---

> > > > > ### Comment · Area_Chair1 · 2018-12-10
> > > > > **And the complement entropy was negated, right?**
> > > > >
> > > > > Also, since the objective is to minimize cross-entropy and maximize complement entropy, I assume that when you tested the unified objective you actually negated the complement entropy term.

---

> > > > > > ### Author Response · Authors · 2018-12-11
> > > > > > **Response to Area Chair1**
> > > > > >
> > > > > > You are totally right. We did negate the complement entropy term (and added it to the primary objective) for maximizing complement entropy. We are sorry about the confusion and we will update the final manuscript to make this more clear: minimizing cross-entropy and maximizing complement entropy (e.g., in Algorithm 1).

---

> > > > > ### Author Response · Authors · 2018-12-11
> > > > > **Response to Area Chair1**
> > > > >
> > > > > Thanks for the comment. We summed the cross-entropy with the normalized complement entropy (Eq.3), and the corresponding advantages were discussed in Section 3.1.

---

> ### Author Response · Authors · 2018-11-14
> **Response to AnonReviewer1 [1/2]**
>
> We would like to thank the reviewer for all the insightful feedbacks. Below we provide the explanations for each question or comment raised by the reviewer:
>
>
> (Q1) How is this idea related to regularization? If we increase the regularization parameter, we can attain sparse parameter vectors.
>
> (A1) Conventionally, regularization techniques (e.g., Ridge or Lasso) are applied on the parameter space. We want to point out that all the results reported in the manuscript, for both baselines and models trained by COT, have already used L2-norm regularization on the parameter space, exactly as specified in the original papers  (e.g., ResNet [1], WideResNet [2], and DenseNet [3]). In other words, COT is applied on top of the existent of those regularization techniques.
>
> If your questions haven’t been addressed satisfactorily, please kindly let us know and we will be happy to discuss further.
>
>
> (Q2) Would this method also complement from overfitting?
>
> (A2) Thank you for the comment. We would like to further clarify what you meant by saying “complement from overfitting.” Our interpretation of the question is: whether COT could be used to fight against overfitting. Overfitting means a model fails to generalize, and in our paper we have reported the generalized performance of models trained by COT on the test data, which confirms models trained by COT generalize better. In addition, we also calculate the loss gap "(testing loss - training loss)" and report the results in the following table, where a smaller gap indicates that a model generalizes better. Experimental results confirm that models trained by COT seem to generalize better due to the smaller gap between training and testing loss.
>
>  "(Testing loss - training loss)” from the state-of-the-art architectures on Cifar10
> ==================================================
> 				                Baseline	        COT
> ResNet-110 			        0.36                 0.33
> PreAct ResNet-18                 0.28                 0.26
> ResNeXt-29 (2×64d)             0.20                 0.19
> WideResNet-28-10		0.23                 0.21
> DenseNet-BC-121           	0.22                 0.22
> =================================================
>
>
> (Q3) In the numerical experiments, the comparison is carried out against a "baseline" method. Do the authors use regularization with these baseline methods? I believe the comparison will be fair if the regularization option is turned on for the baseline methods.
>
> (A3) Yes, the regularization (e.g., L2 Norm) techniques are used in all of the baseline methods, as specified in their original papers (e.g., ResNet [1], WideResNet [2], and DenseNet [3]). We agree with the reviewer that “the comparison will be fair if the regularization option is turned on for the baseline methods,” and that is exactly we did in our paper: all the hyper-parameters, regularization and other training techniques are configured in the same way as in the original papers. For the details of experimental setup, please refer to the Section 3.2 in our manuscript.

---

> > ### Comment · AnonReviewer1 · 2018-11-23
> > **Comparison against regularization**
> >
> > I understand. Unfortunately, the loss gap values in the table do not say much. I apologize for my typo "complement from overfitting." It should be "complement overfitting." To clarify my question, I wonder whether COT can be considered as a complementary or an alternative option against overfitting?

---

> > > ### Author Response · Authors · 2018-11-27
> > > **Follow-up**
> > >
> > > Thanks for your clarification. Based on all of the experiment results we have so far, such as loss gap values, we are only able to claim that models trained by COT generalize better (i.e., better performance on separate test sets).  While achieving better performance on separate test sets is a good indicator that COT does not produce models that overfit, further experiments and theoretical investigations on whether COT can be a rigorous option to guard against overfitting is left as a future work.

---

### Public Comment · (anonymous) · 2018-11-14
**Adversarial robustness claim is highly misleading**

This paper argues in the abstract that "we also show that models trained with both primary and complement objectives are more robust to adversarial attacks."

However, in the evaluation section, the authors only attempt a very simple transferability attack: generate adversarial examples on one model, and transfer them to another. This does not imply adversarial robustness, neither in the white-box nor black-box setting.

To argue black-box robustness, the authors should evaluate against more recent black-box attacks such as the Boundary Attack (ICLR'18) or SPSA (ICML'18). Both of these attacks have effectively broken many black-box defenses in the past.

If the authors wish wish to argue full white-box adversarial robustness, they should further try optimization based attacks (Madry et al. 2018, Carlini & Wagner 2017).

As is, this paper should not claim robustness to adversarial examples: at best, it can claim a 10% improvement in accuracy to transfer attacks.

---

> ### Author Response · Authors · 2018-11-18
> **Thank you for the comments**
>
> Thank you for your comments. We understand that the adversarial attack techniques used here may not be state-of-the-art methods; however, we want to emphasize that the primary goal of this paper is to improve model's accuracy, although experimental results do show that robustness is also one of the benefits of the models trained by COT.
>
> We agree with the reviewer that transfer adversarial attack is different from the classic settings of adversarial attacks. To verify our method under standard adversarial attacks, we have conducted additional experiments on white-box attack, and provided the results below; the experimental results confirmed that COT is indeed more robust to this type of attacks, and therefore we believe the main conclusion that COT is more robust (compared to baselines) to adversarial attack still holds. We will add these results of the white-box attack into the final version of the paper. Additionally, we will rename the current experiments to “transfer attacks” to avoid confusions. The definition of the transfer attacks can be found in several recent publications [1, 2, 3].
>
> For the white-box attacks, we conducted the experiments as also suggested by AnonReviewer3. The update is to set adversarial perturbations to be Epsilon * Sign (Primary gradient + Complement gradient). Results indicate that COT is more robust to this type of white-box attacks under standard settings.
>
> Test errors on Cifar10 under FGSM white-box adversarial attacks
> ===========================================================
> 				              Baseline	                        COT
> ResNet-110 			      62.23%                 		52.72%
> PreAct ResNet-18               65.60%                 		56.17%
> ResNeXt-29 (2×64d)           70.24%                 		61.55%
> WideResNet-28-10	       59.39%                 		55.53%
> DenseNet-BC-121               65.97%                 		55.99%
> ===========================================================
>
> The reviewer also suggested to try out several recent methods on white-box and black-box attacks. We do agree with the reviewer that it's a great idea. However, since the main focus of the current paper is to improve accuracy, and the manuscript is already close to the page limit, we feel it's better to study this problem in a separate paper. As a matter of fact, we are planning on a follow-up work with the focus on the robustness of the models trained with COT.
>
> [1] Nicolas Papernot, Patrick McDaniel, Ian Goodfellow. “Transferability in Machine Learning: from Phenomena to Black-Box Attacks using Adversarial Samples.” Arxiv, 2016
>
> [2] Yanpei Liu, Xinyun Chen, Chang Liu, Dawn Song. “Delving into Transferable Adversarial Examples and Black-box Attacks.” In International Conference on Learning Representation, 2017.
>
> [3] Wieland Brendel, Jonas Rauber, Matthias Bethge. “Decision-Based Adversarial Attacks: Reliable Attacks Against Black-Box Machine Learning Models.” In International Conference on Learning Representation, 2018.

---

> > ### Public Comment · (anonymous) · 2018-11-19
> > **FGSM results are not strong attacks**
> >
> > I understand this is not the main purpose of your paper, but again, you claim "we also show that models trained with both primary and complement objectives are more robust to adversarial attacks." At present, you simply have not shown that fact.
> >
> > Thank you for running some white-box numbers, but FGSM is unfortunately not sufficient. I hate to appeal to authority, to argue this, but see ( https://openreview.net/forum?id=SkgVRiC9Km&noteId=rkxYnt8JpQ&noteId=rkxYnt8JpQ ).
> >
> > Prior work, and papers under submission this year, make very careful claims with respect to adversarial examples.  See for example the Manifold Mixup paper under submission this year that instead writes the correct and honest statement "Manifold Mixup achieves ... robustness to single-step adversarial attacks". You should claim only what you can demonstrate.
> >
> > It is perfectly fine that you want to only show adversarial robustness as a side-effect of your main work, but you should be accurate in how you phrase what you have shown. There is a big difference between being robust to single-step attacks and transfer attacks, and actually being robust. Hundreds of papers claim the former, very few claim the latter.

---

> > > ### Public Comment · (anonymous) · 2018-11-19
> > > **How to talk about FGSM Results**
> > >
> > > I agree with the Nov/19 anonymous comment, and one thing that I'll add is that I think it's worth discussing robustness to the FGSM attack, because it means that the decision boundary is being moved away from the data points, in a certain subset of directions.  I think this is different from adversarial robustness in general, which considers perturbations which give maximum error.
> > >
> > > It would be interesting to think about something like "the volume of the subset of the epsilon-ball around the data points which increases error by k%" - and then we could claim that some methods reduce that volume without claiming that every single point in the epsilon-ball has low error.

---

> > > ### Author Response · Authors · 2018-11-20
> > > **Follow-up**
> > >
> > > Thank you for the clarifications on the recent research trending in adversarial attacks as well as your great suggestions on making the claim precise. We will adopt your suggestion and make it clear in the paper that the proposed training objectives make the models more robust to single-step adversarial attacks instead of claiming general robustness. We will use this new statement consistently across our updated version of the paper.

---

### Author Response · Authors · 2018-11-26
**Manuscript Updated**

We thank all reviewers and the anonymous for the constructive comments. We have updated the manuscript in Abstract, Section 2, Section 3.4, Conclusion and Appendix A to address your feedback and concerns. Here we provide a summary of these updates:

(1) For AnonReviewer3’s main suggestion of forming adversarial attacks using “both” gradients from both primary and complement objectives, we have designed and conducted the additional FGSM (single-step) white-box experiments. The experiments set adversarial perturbations to be generated based on the sum of the primary gradient and the complement gradient (i.e., the gradient calculated from complement objective), while the results indicate that COT is more robust to single-step adversarial attacks under standard settings [1].

(2) To provide more precise claim, we update the original claim “robustness to adversarial attacks” into “robustness to single-step adversarial attacks” according to (1). Additionally, more details of the original transfer attack experiments are provided in the manuscript.

(3) We have added a description about the increase of training time and corrected typos pointed out by the reviewers in the manuscript.

[1] Ian Goodfellow, Jonathon Shlens, and Christian Szegedy. “Explaining and harnessing adversarial examples”. In ICLR’15.

---

### Meta-Review · Area_Chair1 · 2018-12-13
**Novel training objective for deep learning with strong empirical results**

**Confidence:** 5
**Recommendation:** Accept (Poster)

**Metareview:**

This paper proposes adding a second objective to the training of neural network classifiers that aims to make the distribution over incorrect labels as flat as possible for each training sample. The authors describe this as "maximizing the complement entropy." Rather than adding the cross-entropy objective and the (negative) complement entropy term (since the complement entropy should be maximized while the cross-entropy is minimized), this paper proposes an alternating optimization framework in which first a step is taken to reduce the cross-entropy, then a step is taken to maximize the complement entropy. Extensive experiments on image classification (CIFAR-10, CIFAR-100, SVHN, Tiny Imagenet, and Imagenet), neural machine translation (IWSLT 2015 English-Vietnamese task), and small-vocabulary isolated-word recognition (Google Commands), show that the proposed two-objective approach outperforms training only to minimize cross-entropy. Experiments on CIFAR-10 also show that models trained in this framework have somewhat better resistance to single-step adversarial attacks. Concerns about the presentation of the adversarial attack experiments were raised by anonymous commenters and one of the reviewers, but these concerns were addressed in the revision and discussion. The primary remaining concern is a lack of any theoretical guarantees that the alternating optimization converges, but the strong empirical results compensate for this problem.